# Tourism Vulnerability Amid the Pandemic Crisis: Impacts and Implications for Rebuilding Resilience of a Local Tourism System in Vietnam

**Da Van Huynh [1,\*], Long Hai Duong [2,\*], Nhan Trong Nguyen [1] and Thuy Thi Kim Truong [1]**

[1] School of Social Sciences and Humanities, Can Tho University, Can Tho 900000, Vietnam
[2] Department of Global Hospitality and Tourism Management, Kyung Hee University, Seoul 130-701, Korea
\* Correspondence: hvda@ctu.edu.vn (D.V.H.); duonghailong85@khu.ac.kr (L.H.D.)

**Abstract:** Despite the devastating impact of the COVID-19 crisis on the global tourism industry, a few countries have attempted to foster their local tourism economies' recovery by offering distinctive mechanisms which facilitate their safe tourism destinations to restart domestic tourism operations during the pandemic waves. However, there has been little research investigating how different sectors of a local tourism system, particularly in a developing country, seriously suffer from the pandemic crisis but gain encouraging revitalization from the pandemic shocks. Therefore, this study employed Can Tho city as a case study to examine the holistic impact of COVID-19 on different sectors of the local tourism industry and explore the key factors/players contributing to the resilience empowerment and adaptive recovery of the local tourism system. As such, a semi-structured interview approach was employed in this study to collect both quantitative and qualitative data. The study recruited 40 representatives of tourism-related authorities at different levels and 280 managers of different tourism sectors in the local tourism industry. The findings generally reveal the disastrous impacts of the pandemic on the local tourism industry across all tourism sectors but show an unexpected recovery of tourism businesses during the pandemic crisis. The integrated findings also highlight the pivotal role of local governments in crisis governance and destination recovery support during and after the pandemic waves. Similarly, the proactive engagement of local enterprises was found critical toward rebuilding their organizational resilience, and such adaptive transformations were essential for tourism business recovery in new normal conditions. The economic policy response and effective financial schemes were common expected measures toward the tourism industry's recovery in the post-pandemic crisis.

**Keywords:** COVID-19; Can Tho city; resilience; mitigation; Vietnam

## 1. Introduction

The first case of COVID-19 infection in Vietnam was detected on 23 January 2020 (Giang 2022). The tourism activities in Vietnam had almost fallen into a state of crisis and total paralysis until March 2021. Most of the services and activities associated with tourism and hospitality were paused or interrupted, which led to increasingly devastating consequences, including a huge loss of revenue for the tourism industry and jobs for tourism and hospitality workers. From June 2020, the domestic tourism industry started to recover, but with the impact of the other waves of COVID-19, the situation began to struggle again and was projected to face a much more serious challenge to recover post-pandemic.

According to the International Monetary Fund (IMF), some economic sectors in Vietnam were found to be severely affected, especially tourism, transportation, and accommodation services (Kim Thoa 2020). Before the COVID-19 pandemic, Vietnam's tourism was estimated to welcome 18 million international visitors and serve 85 million domestic

visitors, which significantly contributed around USD 32.8 billion (equivalent to 9.2% of the GDP) (Huy 2021). Yet, the number of international visitors to Vietnam in 2020 sharply decreased by 99% compared to that of the previous year, which revealed an enormous pandemic impact on the national tourism industry (General Statistics of Vietnam 2021).

However, Vietnam has attempted to save the tourism industry by fostering the domestic tourism economy given the adverse standstill of the global tourism industry. Apart from launching multiple stimulus packages for domestic tourism businesses, the central government also allowed potential tourism destinations to restart their activities as long as the epidemic control regulations were highly committed. As such, tourism destinations with limited or no Coronavirus cases were encouraged to tailor their pandemic prevention protocols in a way that could stimulate the tourism recovery of local tourism sectors and tourists' travel demand under the COVID-19 situation. Despite the pressure of the lingering impact of the pandemic, a few tourism destinations resumed up to 60% of their destination capacity over the first and second waves of the epidemic crisis (Huynh et al. 2022). In this regard, the local tourism industry could be used as an effective temporary tool to gain the economic revitalization of the tourism industry; thereby, socio-economic pressures could be alleviated (Ozili 2020).

Despite the encouraging recovery of a few local tourism systems amid COVID-19, there has been a lack of understanding of how a local tourism industry rebuilds its resilience and recovery during the pandemic waves. Therefore, this study aims to investigate how the key sectors of a local tourism system suffer from and adapt well to the pandemic's challenges. To shed light on this research phenomenon, this study examines key questions as follows:

1. To what extent does the COVID-19 crisis affect the key sectors of a local tourism system?
2. What are the key factors influencing the destination's resilience and recovery amid the pandemic crisis?

## 2. Literature Review

The current global pandemics have generated a huge impact on all aspects of human life. The proliferation of global pandemics might stem from a plethora of causes, including increasing communication and global mobility, urbanization, the high concentration of people, and the growth of the global transportation system (Pongsiri et al. 2009). According to Qiu et al. (2017), there were only three global pandemics during the 20th century but four pandemics within 20 years of the following century. The World Bank warned that the dark prospect of COVID-19 pandemic impacts and other potential disasters might seriously affect people's lives and our global sustainable development goals until 2030 (McKibbin and Fernando 2021). More seriously, developing and under-developed countries might face bigger challenges to recover their tourism economies since the pandemic could be lingering and unprecedented (Peasah et al. 2013).

Academic research regarding the assessment of pandemic impacts on tourism has received the attention of scholars around the world for over 20 years. The world has recorded global epidemics that somewhat affected the supply and demand of the tourism industry. Evidently, despite a far smaller impact than that of COVID-19, such disease outbreaks as SARS in 2003 or H1N1 in 2009 have affected the growth of the tourism industry, but these effects were likely to be short-term impacts (Gössling et al. 2020). Richter (2003) claimed that public health problems influenced tourism activities in both host and guest countries. This also means that countries share both benefits and challenges when experiencing a global health crisis. A typical example of this is the decreasing number of tourist arrivals in Hong Kong during the SARS epidemic in 2003, which has revealed its negative impact on the host tourism industry despite its small-scale influence (Siu and Wong 2004). More seriously, the swine flu epidemic in 2009 hit Mexico's tourism quite heavily with a drop of about 1 million visitors and a loss of USD 2.8 billion in revenue (Rassy and Smith



2013). The UNWTO has assessed the impact of several global events and their impact on the tourism industry (UNWTO 2020). Gössling et al. (2020) also pointed out that the tourism industry is a recoverable sector from external impacts, but he also claimed that the tourism economies might face a by far devastating impact of the COVID-19 crisis. According to the World Tourism Organization's prediction in April 2020, even if the COVID-19 situation was more positive by mid-2020, the world tourism industry still lost 20–30%, with a decline from USD 290 to 440 billion. In case the COVID-19 situation becomes more complicated in the near future, the world tourism industry might suffer a huge loss of its revenue due to the lingering standstill of the global mobility restrictions (UNWTO 2020). Unfortunately, the situation of the COVID-19 pandemic became worse in 2020, which led to an enormous economic loss of around 80% of its revenues (UNWTO 2021).

In May 2020, the World Tourism Organization released a recommendation to take action by local authorities and governments against the COVID-19 epidemic, proposing how the tourism industry could respond in a more effective and resilient way. Such recommendations from the World Tourism Organization included 23 solutions divided into three categories, involving better controlling diseases and minimizing risks, stimulating and accelerating recovery, and being well-prepared for the future (UNWTO 2020). There are three main fields of academic research regarding the impact of epidemics on tourism, including the impact of disease on tourism demand (Kuo et al. 2008; Page et al. 2012; Solarin 2015), the tourism industry's response to pandemics at a local and national level (Cooper 2006; Maphanga and Henama 2019; Huang and Zeng 2022), and the influence of pandemics on specific aspects of the tourism industry, such as transportation, restaurants-hotels, event tourism, and so on (Henderson and Ng 2004; Le and Phi 2021). Although there have been many studies on the impact of disease on tourism on a regional or national scale, there is a lack of systematic research on the impact of the pandemic crisis on different sectors of the local tourism system.

Despite a short outbreak of the pandemic, the enormous impact of the COVID-19 epidemic on the world tourism industry has gained increasing attention within academia. Initial efforts from the scholars to explore the vulnerability and strategic adaptation for the global tourism industry recovery (Gössling et al. 2020). Scholars pointed out that COVID-19 has had a strong influence across all sectors of the tourism industry (e.g., aviation, accommodation-restaurant, sport and event travel, and yachts) and the world tourism industry might see USD 2.1 trillion in losses in 2020. The aviation industry suffered heavy losses with more than USD 250 billion of decline compared to that in 2019. This study has highlighted the long-term impacts of the COVID-19 pandemic on the world tourism industry and the fundamental change in the global tourism industry, including reducing the dynamics, demand, and scale of the tourism industry and changing tourist habits, which strongly affects the tourism enterprise community.

The United Nations Development Program (UNDP) has released a report titled "A Conceptual Framework for Analyzing the Economic Impact of COVID-19 and its Policy Implications" (Hevia and Neumeyer 2020) in which the evaluation frameworks of COVID-19's direct and indirect effects on the global economy were proposed. In addition, the report by the World Tourism Organization, namely "UNWTO World Tourism Barometer May 2020, special focus on the Impact of COVID-19" (UNWTO 2020), conducted a fairly specific SWOT analysis of strengths and weaknesses to identify the opportunities and challenges of the tourism industry against the COVID-19 pandemic. This report identified the impacts of COVID-19 on the tourism industry, including an increase in unemployment; SME bankruptcy; workers' income reduction; and increased skepticism and fear of tourists when deciding to travel. These key concerns have led to a decline in tourism demand and difficulties for the tourism supply chain. In addition, the studies of (Sigala 2020) and Isaac and Keijzer (2021) also stressed the degree and extent of the impact of the COVID-19 epidemic on the tourism industry.

Other papers investigating the impact of COVID-19 on Chinese tourism by Hoque et al. (2020) and India's tourism industry by Kasare (2020) indicated the huge and long-term

pandemic impact on both domestic and international tourism in these countries. Similarly, other studies also highlighted the strong impact of the COVID-19 pandemic on Malaysia's aviation and restaurant-hotel sector (Karim et al. 2020), tourism in small islands in Indonesia (Dinarto et al. 2020), and online skating tourism in Austria (Correa-Martínez et al. 2020). Such common concerns regarding unemployment, bankruptcy, revenue loss, and budget deficit were serious consequences brought about by the COVID-19 pandemic (Kasare 2020). For tourism-dependent countries, the economic impact might be much more serious since all tourism sectors might suffer from the lingering crisis (Centeno and Marquez 2020). A study by Bakar and Rosbi (2020) also predicted that world tourism might collapse under the impact of COVID-19 if there were no proper measures to rebound tourism businesses. Recently, there have been several publications dealing with world tourism prospects in the post-COVID-19 period. Chang et al. mentioned the perspective of the sustainable development of the world tourism industry in the post-COVID-19 period through an analysis of advantages and disadvantages. The authors suggested that there should be an alternative approach to tourism recovery in short term.

In Vietnam, most of the research results on the impact of COVID-19 mainly refer to the health sector, border protection, and the economy and society in general. Some earlier publications in Vietnam in specialized journals and reports mentioning the impact of and solutions to recovering from the COVID-19 epidemic typically include research by Phạm et al. (2020a, 2020b) exploring the impact of the COVID-19 epidemic on tourism at local levels. The following studies (Quang et al. 2022; Van Nguyen et al. 2020) assessed the short- and long-term challenges of COVID-19 to the tourism business and its response policies, as well as recommendations for firm support to accelerate recovery. The study outlined the scenarios of COVID-19 impact on Vietnam's tourism industry in the future. The study also proposed specific strategies and solutions to restore Vietnam's tourism after the COVID-19 pandemic's impact.

In April 2020, the Vietnam Tourism Advisory Council surveyed 394 tourism and tourism businesses to assess the impact of the pre-COVID-19 "freezing" of the tourism industry and proposed several recovery solutions based on the synthesis of suggestions from businesses and experts. Thereby, Vietnamese government policies have been implemented based on the suggestions of the World Tourism Organization and tourism experts. Accordingly, the financial policies have been adjusted in a way that could generate positive impacts on the tourism industry revitalization. More specifically, several financial measures have been implemented to ease the financial crisis facing the tourism industry (e.g., prioritizing tax exemption or lowering bank interest rates). These policy-related innovations not only contribute to stimulating the tourism sector's recovery during the pandemic crisis but also support the vast majority of unemployed workers dealing with their temporary financial pressure. Moreover, the local governments were empowered by providing them with special mechanisms to save their local tourism systems. As a result, several tourism destinations that could ensure the epidemic control regulations were facilitated to restart domestic tourism activities and gain encouraging recovery in the post-pandemic crisis. Last but not least, local tourism systems have actively adapted to the pandemic impact by taking innovative transformations at the organizational levels, which strengthens their resilience against the lingering crisis.

Despite the encouraging recovery of a few tourism destinations during the pandemic, there is still a lack of research investigating the holistic vulnerability of local tourism systems and how different sectors of a local tourism industry rebuild their resilience amid the pandemic shocks. Therefore, this study aims to provide insight into the vulnerability of a local tourism system across key sectors and its strategic implications for tourism destination resilience and recovery in the post-pandemic crisis.

## 3. Research Methodology

This study used a semi-structured interview approach to examine the vulnerability of a local tourism system against the pandemic impact and how different sectors rebuild their resilience and recovery strategies in the post-pandemic crisis.

Regarding the case study selection, this study employed Can Tho city to shed light on the research phenomenon for some critical reasons. First of all, this tourism destination has been hit hard during the initial waves of the pandemic. Despite the enormous impact of COVID-19 waves on the local tourism system, Can Tho was one of the few typical destinations reviving around 60% of its tourism capacity (Huynh et al. 2022). Importantly, this tourism destination was offered special mechanisms and policy initiatives by the local government, which fosters the local tourism system to resume tourism activities during the pandemic outbreaks. Noticeably, the encouraging recovery of different tourism sectors provide valuable lessons regarding how a local tourism could adapt well and recover from the long-lasting pandemic crisis.

To gain insight into the vulnerability and adaptive recovery of the local tourism destination from the perspective of local governments, this study targeted the potential respondents who were in charge of tourism crisis management and governance. This enabled the researchers to explore how local governments balanced the dual targets between pandemic control and tourism revitalization. More specifically, this study recruited 40 officials from many tourism-related organizations, such as the Tourism Association, Department of Culture, Sports and Tourism, Center for Tourism Development, Department of Information and Communications, Department of Health, Department of Planning and Investment, Department of Finance, and local government at the district level. Creswell and Poth (2016) suggested that a minimum number of participants for a semi-structured interview show be at least 25; therefore, the number of informants selected in this study met the requirements of data collection. The interviews were conducted between October and November 2020.

The main interviews focused on the impact level of the COVID-19 pandemic on the key sectors of the local tourism system. The interviews focused on exploring the resilience of the tourism industry, measures to limit the negative impact of the epidemic on the tourism industry, measures to restore the tourism industry, measures to prevent anti-epidemic, tourism marketing, and promotions to revive the tourism industry in Can Tho city. As such, 280 respondents who were managers of different sectors of the local tourism system (e.g., hotels, restaurants, tourist attractions, and tour operators) were recruited for the interviews.

Regarding the respondent selection rationale, the random informants recruited in this study should be in charge of executive responsibilities, which might ensure that the respondents could provide profound data to meet the research objectives. In addition, the respondents should be managers of different tourism sectors of the local tourism system. Importantly, local tourism sectors selected in this study should be vulnerable businesses but successfully recovered from the pandemic impact. Last but not least, tourism enterprises selected in this study should be located in different areas of the examined case study in order to reflect the general status of the local tourism system. These critical criteria allow the researchers to shed light on the research phenomenon (see Table 1).

**Table 1.** Structure of quantitative research samples.

| Item | Number of Interviews | Sample Rate (%) | Overall | Percentage (%) |
|---|---|---|---|---|
| Accommodations | 80 | 28.5 | 275 | 29.1 |
| Travel agencies | 50 | 17.9 | 59 | 84.7 |
| Dining establishments | 50 | 17.9 | 200 | 25.0 |
| Tourism fruit orchards and homestays | 30 | 10.7 | 41 | 73.2 |
| Tourist attractions | 70 | 25 | 70 | 100 |

Source: Data from Huynh et al. (2021).

A total number of 280 respondents who were managers of hospitality-and-tourism enterprises were recruited for the semi-structured interviews. A thematic analysis was employed to interpret the qualitative data, while the quantitative data were analyzed by using SPSS 20 software.

## 4. Research Findings and Analysis

### 4.1. The Decline in Customers, Revenue, Service Utilization, and Staff Due to the COVID-19 Pandemic

The research findings generally reflect a significant downward trend in tourist arrivals, revenue, and service consumption but increasing employment downsizing in the research site during the waves of the COVID-19 pandemic.

The quantitative data of 280 managers of accommodation, dining establishments, travel agencies, and tourism sites revealed that the pandemic influenced a wide range of businesses in the research sites in a negative way. The findings indicated that the proportion of the impact of the pandemic on local businesses was at a serious and very serious level, with 49.3% and 40%, respectively. Reportedly, there was a significant decrease in the number of customers, sales, services, destination capacity, tourism assets, and staff in the tourism industry. More specifically, there was a decline in each enterprise in terms of customers, revenue, and employees by 11,234.4 people, VND 2.6 billion, and 9.3 employees, respectively, with relative percentages of 62.3%, 61.3%, and 50.3%, respectively. The decrease in customers also led to a decline in the capacity of exploiting and using services and assets at an average level of 57.4% per item. The quantitative data strongly reconfirmed the above concerns with over 63% of enterprises suspending operations, 7% of enterprises dissolving out of total suspended enterprises, 50% of enterprises temporarily shutting down, over 70% of employees being affected and unemployed or changing jobs, only 16% of enterprises being able to maintain business operations, and 14% cutting down on scale and staff to continue operating.

The results of qualitative interviews with many stakeholders related to tourism management and development also indicated a bleak picture for the tourism industry in Can Tho city during the COVID-19 epidemic. Such words and phrases as "collapse", "extremely heavy losses", "bankruptcy", "job loss", and "create a burden on society" were found common to indicate the serious impact of the epidemic on tourism in Can Tho city.

One of the typical assessments of the impacts of the COVID-19 epidemic on the tourism industry of Can Tho city was stressed:

> "*The epidemic has reduced the number of visitors, revenue, and human resources. and worse still, dissolve many businesses. This causes pressure on the economy, society and serious damage to the local tourism industry*" (a representative of Cai Rang District).

### 4.2. Measures to Minimize the Negative Impacts of the COVID-19 Pandemic on the Tourism Industry

Despite the unexpected crisis due to the pandemic, the research respondents positively claimed that the negative impact of diseases on tourism could be mitigated through a variety of measures, including limiting the spread of diseases in the tourism environment and implementing strategies to alleviate negative impacts. In addition, the research participants also believed that the policies and support from local authorities to some extent contributed to controlling and alleviating negative effects. One of the respondents stressed the importance of limiting the spread of disease in the tourism environment by quickly controlling the disease outbreak and thereby returning the business to normal and reducing negative impacts.

In addition to the role of the local authorities' assistance, all of the managers participating in the interview believed that they also played a pivotal role in the success of local epidemic prevention by advising their employees and customers to wear masks, wash hands with an antiseptic solution, and keep contact distance. Further, tourism enterprises

also claimed that they were committed to regularly cleaning their workplace or even agreeing on business suspension to prevent an outbreak with a high rate of respondents, with 85% and 61.4%, respectively. Measures such as serving clients only in non-epidemic locations and other options (e.g., online work, staff training in epidemic prevention knowledge and skills, and compliance with epidemic prevention recommendations from The Ministry of Health and local authorities) were also prioritized by the respondents, with 43.6% and 8.6%, respectively (Table 2).

**Table 2.** Measures to limit the spread of diseases in the tourism businesses' environments (n = 280).

| Measure | Percentage (%) |
| --- | --- |
| Advising employees and customers to prevent and combat epidemics | 95 |
| Ensuring regular hygiene at business establishments | 85 |
| Serving customers only in locations without the pandemic | 43.6 |
| Contemporary business suspension | 61.4 |
| Other measures | 8.6 |

Source: Data from Huynh et al. (2021).

To promote the domestic tourist market, the local authorities have supported the local tourism enterprises by offering prioritized business loans, which allows the tourism sectors to restructure their business operation in the post-pandemic waves. In addition to tourism management and development agencies, enterprises and destinations prioritized employee safety and the domestic market, with 72.9% and 71.4%, respectively, whereas other measures regarding saving water and energy, resigning employees, and reducing staff salaries were fewer favorite measures, with 55%, 32.9%, and 17.1%, respectively (see Table 3).

**Table 3.** Measures to minimize the negative impacts of the pandemic on tourism business activities of local enterprises (n = 280).

| Measure(s) | Percentage (%) |
| --- | --- |
| Focus on exploiting the domestic market | 71.4 |
| Ensure health safety for employees | 72.9 |
| Reducing staff salaries | 17.1 |
| Employee downsizing | 32.9 |
| Saving water and energy resources | 55 |
| Other measures | 7.9 |

Source: Data from Huynh et al. (2021).

All of the tourism managers agreed that the epidemic was complicated and might have a long-term impact on the tourism industry. However, they all agreed that the group of pandemic prevention measures at the local to central level should be maintained and strengthened, while domestic tourism maintenance should be taken into account to sustain tourism-related businesses to a certain degree.

The research participants also agreed that effective policies or support from local authorities played a key role in supporting local businesses and tourism destinations to minimize the negative effects of the pandemic on businesses. According to the respondents, before the survey was conducted, the local authorities had initiated policies and support for businesses and destinations such as improving business capacity for businesses (31.8%), reducing tax (21.4%), a loan with a preferential interest rate (14.3%), and tax exemption (10.4%). In addition to the above policies and supports, the authorities also reduced electricity and water prices, boosted tourism marketing and promotion, and dealt with the unemployment of employees. However, it is likely that the overall support has not met the needs of the tourism and hospitality sector (Table 4).

**Table 4.** Local authorities' policy and support (n = 280).

| Measure | Percentage (%) |
|---|---|
| Tax reduction | 21.4 |
| Loan with a preferential interest rate | 14.3 |
| Tax exemption | 10.4 |
| Enhancing business capacity for enterprises | 31.8 |
| Policy/other support | 20 |

Source: Data from Huynh et al. (2021).

*4.3. Measures to Revive the Tourism Industry Due to the Impact of the COVID-19 Epidemic*

The COVID-19 epidemic is an unprecedented tragedy for mankind. The complicated development of the disease caused a lot of damage to the economy, society in general, and the tourism business in particular. The tourism industry of Can Tho city has experienced the third wave of epidemics with many difficulties and challenges (the first time on 1 April 2020, the second time on 25 July 2020, and the third time on 27 January 2021). To adapt well to the pandemic challenges, several businesses and destinations in Can Tho city have taken measures such as creating trust for customers (66.4%), promoting the communication of products (51.1%), reducing service prices (50.7%), and offering promotions (50%). In addition, other measures have also been used to revive the tourism business such as developing new products and seeking new business opportunities (Table 5).

**Table 5.** Measures to recover business operations of enterprises (n=280).

| Measure(s) | Percentage (%) |
|---|---|
| Service discounts | 50.7 |
| Promotion | 50 |
| Promoting communication | 51.1 |
| Empowering trust for customers | 66.4 |
| Other measures | 7.1 |

Source: Data from Huynh et al. (2021).

The above measures have positively contributed to the recovery of the tourism business in Can Tho city. According to the assessment of many tourism management and development agencies, the resilience of the tourism industry in Can Tho city in both the short and long term is promising:

*"By the end of 2020, the domestic tourism industry will be able to recover quickly. The coming year 2021 will mark the return and recovery of Can Tho's tourism industry"* (Mekong Delta Tourism Association).

According to the respondents, businesses and tourist destinations should implement a wide range of measures that should be flexible, sufficient, and effective to meet the needs of each local community.

## 5. Discussion and Conclusions

This study has highlighted the enormous impact of the COVID-19 pandemic on the local tourism system across different sectors of the economy. To alleviate the vulnerability of local tourism businesses, local governments were found to play a pivotal role in a local tourism system's recovery. Essentially, the financial support for tourism enterprises was indispensable to resume their tourism operation when most of the tourism businesses suffered from the long-lasting crisis, especially small and medium enterprises. More specifically, the monetary promotion measure indicates its decisive role in restructuring the destination image, ensuring the quality of tourism products and services and stimulating the domestic market demand. Although this is considered a temporary solution, a local tourism industry could be resumed systematically only when all tourism sectors could have

access to the financial support scheme. This study reconfirms a previous finding by Duong et al. (2022) that financial stimulus packages might be very crucial for not only the post-crisis recovery of tourism sectors but also for stimulating tourists' travel demand in post-pandemic destination recovery.

The adaptive strategy to co-live with COVID-19 enables the local tourism system to endeavor to alleviate the impact on the most vulnerable sectors and seek alternative approaches to recover in a more resilient way. In this sense, it requires all local tourism enterprises to work together to rebuild their organizational resilience to better respond to the crisis. This is critical to the recovery of the tourism system since most tourism sector businesses are interrelated (Huynh et al. 2021). Despite the necessity of the local government's support, the findings in this study imply the importance of proactive adaptation at the organizational level, which may enable tourism enterprises to ensure the quantity and quality of post-pandemic service delivery.

The majority of the examined respondents in this study stressed that the visionary mindset of "co-living with the pandemic" has fostered their business in a more sustainable way. The findings in this study indicate that enterprises with well-prepared strategies at the beginning are more proactive and effective in coping with the long-lasting pandemic. In addition, collaboration among the tourism sectors in the examined case study was found to be a critical factor that strengthens the organizational resilience and adaptative empowerment of the tourism system during the waves of the pandemic. First of all, the shared benefits during contemporary effective coordination enable different sectors to survive in the short term, which allows the enterprises to restructure their adaptive and transformative strategies to foster efficiency in the long term. This research finding is found similar to that of a study by Cygler et al. (2018) which stressed the importance of coopetitive relationships among the competitors to gain benefits in a contemporary context. It is quite understandable that effective coopetition among tourism businesses during the waves of the COVID-19 pandemic co-creates multi-benefits and economic advantages for the whole tourism system.

Moreover, rebuilding organizational capacity was found critical to the recovery of tourism enterprises. The findings in this study indicate that the majority of enterprises focus on improving human resources by providing training courses to empower employees' professional skills. This enables tourism workers to ensure their operational practice aligns with disease prevention regulations. In this regard, the re-opening of tourism activities could minimize the spreading of the Coronavirus among employees and tourists. The majority of enterprises are committed to sustainable practices. One of the key changes in the mindset of tourism enterprises is the co-living intention with the pandemic. More enterprises are engaged in the post-pandemic recovery of businesses by promoting and marketing tourism programs to stimulate tourism demand recovery.

To be more resilient to the lingering pandemic wave, "thinking outside the box" was found to be critical to business maintenance in both the immediate and long-term recovery of the hospitality industry. The majority of the respondents in this study partially reflected their inactive response to the first wave of the pandemic but were generally more elastic to the following waves. More specifically, the examined hospitality enterprises decided to live with the ongoing COVID-19 pandemic. Whereas the Vietnamese government attempted to prevent the spread of the COVID-19 pandemic by closing some of or the entire economies of regions seriously affected by COVID-19, most of the enterprises have made necessary changes to co-live with the pandemic. As a result, these enterprises appeared to have made reasonable and more resilient decisions and visionary strategies against the ongoing variants of the COVID-19 pandemic. These findings are similar to previous studies (González 2021; Murakami et al. 2021).

From the demand side perspective, digital transformation was not only critical to internal business operation but also essential to meet the new demand of customers who generally preferred convenient service quality and safety-first priority post-COVID-19.

The findings in this study reconfirm the significance of innovative practice to sustain business practice during the pandemic crisis (Huynh et al. 2022; Szabo et al. 2021; Rodríguez-Antón and Alonso-Almeida 2020). Other solutions which were stressed include supporting tourism businesses in the area to revive human resources for the recurrence of future tourism development, actively preventing epidemics, and highlighting energy saving and online business. As highlighted by the representative of business and destination managers in this study, the tourism sectors have to actively engage in the recovery process by rebuilding their organizational adaptation and preparing well for the prospective scenarios of the pandemic waves. The research findings indicate the effectiveness of the early epidemic response, and the engagement of all stakeholders in the communities has enabled the local government to put the epidemic impact under control. Apart from this, the central government's pandemic prevention regulations and tourism enterprises' measures increase the possibility of better control of the pandemic, thereby empowering the tourism resilience of the city. This reconfirms the belief that tourism resilience depends on the involvement of the communities and may quickly recover within a year.

The findings in this study implies that the leading role of local authorities and the necessity of decentralizing power to the local governments allow them to flexibly balance between pandemic control and local tourism revitalization accordingly. This finding reconfirms the importance of effective governance that may determine the epidemic prevention and tourism resilience (Wan et al. 2022). However, economic support and capacity building for tourism businesses may need to be considered on case-by-case basis. To reduce the stamina and increase the resilience of the business operations of enterprises in different vulnerable regions, the locality needs to implement well-supportive policies for enterprises and workers seriously affected by epidemics under the direction of the central government regarding electricity price reduction, capital support, tax support, the suspension of paying social insurance to the retirement and death fund, delaying the time of the payment of trade union fees, and so on.

For businesses, it is critical for enterprises to restructure human resources towards compactness, quality, and efficiency, promote communication and exploit a disease-free market, create trust for customers, develop new products, and offer service discounts and promotions. Minimizing unnecessary costs and taking advantage of the government's support policies are critical for the local businesses. The active involvement of local enterprises indicates their critical role of mitigating the impact on their business and their measures have been effective during the lockdown period. According to Sharma et al. (2021), the tourism enterprises' need to relocate their business operations is an effective measure to overcome the pandemic's impact without governmental support. For major issues regarding unemployment, payment cuts, and company downsizing, economic measures from the local and central governments of Vietnam should be strengthened by increasing financial support to enable local businesses and people to overcome their current difficulties, thereby fostering quicker tourism resilience (Zhang et al. 2021; Williams 2021).

Theoretically, this study has provided some critical contributions toward disaster crisis management. First, this study highlights the leading role of government in disaster management and post-crisis destination recovery. As discussed earlier, proactive preparedness and policy-related measures (e.g., monetary promotion) foster tourism destination revitalization in a more resilient and effective way. At the same time, the adaptive transformation at the organizational level by taking disruptive technologies and changing mindset is also critical to organizational resilience in both the short and long term. This stresses the important role of tourism enterprises as key stakeholders during the pandemic and post-pandemic recovery. Due to the complicated nature of the COVID-19 pandemic, staged crisis management should be flexible to better cope with the unprecedented impact of the global pandemic.

Practically, this study provides several important implications for the tourism industry to better adapt to the current pressure of the pandemic. A quick and timely crisis response at the organizational level may contribute to better epidemic control and maintenance of the local businesses to a degree that mitigates a total crisis in tourism enterprises' businesses. Lessons learned from the failure of the tourism industry during the first wave of the pandemic suggest that tourism enterprises may enhance their resilience by taking innovative measures such as disruptive technologies. Apart from strategic transformation and rebuilding organizational elasticity initiatives, long-term visions such as a coopetition strategy may provide all tourism-related services a better opportunity to survive during and after the pandemic crisis. This also means that sharing benefits among tourism sectors will enhance their survival chance in the post-pandemic recovery. Therefore, the activeness, consensus, and cooperation of all stakeholders may contribute to rebuilding a more resilient tourism system.

Despite the efforts to minimize the limitations in this study, there is more room for future research. Prospective studies may examine the key roles of other stakeholders toward epidemic prevention and destination resiliencies in post-pandemic recovery in other similar contexts. In light of the current findings, potential research may explore factors determining the ability of destination recovery and the extent to which local and central policies may be helpful in the resilience of emerging cities in developing countries.

**Author Contributions:** Conceptualization, D.V.H.; methodology, D.V.H. & L.H.D.; software, N.T.N.; validation, T.T.K.T.; formal analysis, D.V.H. & N.T.N.; investigation, D.V.H. & L.H.D.; resources, D.V.H.; data curation, D.V.H. & N.T.N.; writing—original draft preparation, D.V.H.; writing—review and editing, D.V.H., N.T.N., T.T.K.T. & L.H.D.; visualization, N.T.N.; supervision, D.V.H.; project administration, D.V.H. & T.T.K.T.; funding acquisition, D.V.H. All authors have read and agreed to the published version of the manuscript.

**Funding:** This publication is supported by the Australian Government through the Australian Alumni Grants Fund. The opinions expressed in this publication are those of the authors and do not necessarily reflect the views of the Australian Government.

**Institutional Review Board Statement:** Ethical review and approval were waived for this study since no (sensitive) personal data and information was processed.

**Informed Consent Statement:** Informed consent was obtained from all subjects involved in the study.

**Data Availability Statement:** Data are available upon reasonable request.

**Conflicts of Interest:** The authors declare no conflict of interest.

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
