# Peer review of "Tourism Vulnerability Amid the Pandemic Crisis: Impacts and Implications for Rebuilding Resilience of a Local Tourism System in Vietnam"

_socsci, doi:10.3390/socsci11100441_

Round 1

Reviewer 1 Report

I have some suggestions which can improve your research work, such as

Line number 13 , author (s) should replace the word city with country, please.

The author (s) should mention the statistical technique which they used to draw the results of the study.

The research questions of the study are not clear, please state them clearly in the introduction section. Furthermore, this proposed question must address in the introduction section and as well as in the literature review. Moreover, the methodology of the study must align with a proposed research question.

Furthermore, the author (s) should propose the hypothesis in the literature section which must align with the proposed research questions.

Although, the author should explain the nature of the population, is it known or unknown, and what formula they used to define the sample size. Besides, author (s) how they get a 100% response rate? This is humanely impossible. Furthermore, information regarding questionnaires is missing, from where they have been adopted or adapted or developed etc. The author (s) should attach the questionnaire in the appendix. Moreover, the author (s) has used the random sampling techniques, please give the justification, to why they have taken the simple random technique.

In the respective qualitative technique, the author should give the situation level, which is missing.

In respect of data analysis, the author (s) must use the NVIVO for qualitative analysis and smart pls for quantitative analysis. The data analysis of the study is confusing and unjustifiable.

The same goes for the last section. The author should align the proposed question with the last section, they should give the answer as well as they must discuss the quantitative analysis results.

Author Response

Dear Reviewer 1

We would like to express our sincere appreciation for your time and excellent comments on our paper. We have carefully addressed all the comments as you can see below.

Point 1: Line number 13 , author (s) should replace the word city with country, please.

Response 1: Yes, it has been replaced accordingly. Please see Introduction, paragraph 3.

Point 2: The author (s) should mention the statistical technique which they used to draw the results of the study.

Response 2: Yes, the Research methodology has been modified. Please see Research Methodology, Paragraph 2.

Point 3: The research questions of the study are not clear, please state them clearly in the introduction section. Furthermore, this proposed question must address in the introduction section and as well as in the literature review. Moreover, the methodology of the study must align with a proposed research question.

Response 3: Yes, the research questions have been added to the Introduction section (Please see the last two paragraphs). Accordingly, the Research methodology section was modified (please see paragraph 2)

Point 4: Furthermore, the author (s) should propose the hypothesis in the literature section which must align with the proposed research questions.

Response 4: Yes, the literature review section was modified to align with the research questions. By using the semi-structured interviews, we aim to collect descriptive statistics and in-depth information from the interviewees.

Point 5: Although, the author should explain the nature of the population, is it known or unknown, and what formula they used to define the sample size. Besides, author (s) how they get a 100% response rate? This is humanely impossible. Furthermore, information regarding questionnaires is missing, from where they have been adopted or adapted or developed etc. The author (s) should attach the questionnaire in the appendix. Moreover, the author (s) has used the random sampling techniques, please give the justification, to why they have taken the simple random technique.

Response 5: Yes, we have addressed the points by adding key information about population and explain why our sample is meet the study criteria (please see Research methodology section, paragraph 2). Since we conducted face-to-face interview, most of the information was double checked before the end of interview. Therefore, all responses were totally qualified. Regarding the questionnaire, we will attach it with our revised manuscript as appendix, if possible. Concerning the random sampling technique, since we could not interview more than a thousand of hospitality and tourism enterprises, we encoded an enterprise with a number and randomly pick up one by one for the interview.

Point 6: In the respective qualitative technique, the author should give the situation level, which is missing.

Response 6: Yes, we have modified it in the first paragraph (Research methodology Section)

Point 7: In respect of data analysis, the author (s) must use the NVIVO for qualitative analysis and smart pls for quantitative analysis. The data analysis of the study is confusing and unjustifiable.

Response 7: We totally agree with you that NVIVO will be great as a qualitative analysis tool, but we are comfortable with manual thematic analysis. Because we just aim to do descriptive statistics to reflect a holistic status of the vulnerability of local tourism system against COVID-19, we believe that using Smart PLS will be great for testing hypotheses using PLS SEM which will be interesting for future studies.

Point 8: The same goes for the last section. The author should align the proposed question with the last section, they should give the answer as well as they must discuss the quantitative analysis results.

Response 8: Yes, we have improved the discussion section by revisiting research findings, research questions and literature review.

Many thanks again.

Kind regards

Authors

Reviewer 2 Report

The author might want to include 'Vietnam' or 'Can tho city' in the title perhaps. Emerging city is so vague!

The author might want to consider sending it for language editing

The methodology needs more clarification. why choose mixed methods? Is qualitative alone insufficient? It is unclear whether both quantitative and qualitative are based on simple random sampling or just the questionnaire.

Why are 'new opportunities during pandemic' 'corporate recommendations' asked in the quantitative instead of interviews?

The authors could perhaps show who are the 40 representatives that they interviewed (breakdown in a table) so it is clearer how many are from local government or tourism associations etc.

Why October and November? Is that intentional?

What kind of questions were asked? The author could also share the questions in the appendices  

Findings and analysis/Discussion- The findings presented were very generic (almost sounds like it is applicable to any city). For instance decline in customers, service and staff all this are happening worldwide. Table 2 shows the measures that limit the spread of diseases which were also generic. You don't need an interview to know this. 

The paper will read better if the author is able to provide more information specifically applicable to Can tho city instead of generic findings.

Author Response

Dear Reviewer 2 

We would like to express our sincere appreciation for your time and critical comments on our paper. We have carefully addressed all the comments as you can see below.

Point 1: The author might want to include 'Vietnam' or 'Can tho city' in the title perhaps. Emerging city is so vague!

Response 1: Yes, I agree with your suggestion. The title has been renewed.

Point 2: The author might want to consider sending it for language editing

Response 2: Yes, we agree that the manuscript will be sent for language editing before the final proofreading state.

Point 3: The methodology needs more clarification. why choose mixed methods? Is qualitative alone insufficient? It is unclear whether both quantitative and qualitative are based on simple random sampling or just the questionnaire.

Response 3: Since we also need to collect quantitative data to assess the economic pandemic impacts on the tourism sectors. Therefore, we design the semi-structured interview to collect both quantitative and qualitative data. The random sampling technique was used for recruiting managers of the tourism and hospitality since we could not interview over a thousand of enterprises. We have clarified these points in the Research methodology section.

Point 4: Why are 'new opportunities during pandemic' 'corporate recommendations' asked in the quantitative instead of interviews?

Response 4: since we hope to quantify common measures recommended by the enterprises. I think you are right because we can collect this information by using interviews.

Point 5: The authors could perhaps show who are the 40 representatives that they interviewed (breakdown in a table) so it is clearer how many are from local government or tourism associations etc.

Response 5: Yes, we have added/clarified this point in the Research methodology (please see paragraph 2). To be exact, 40 respondents (e.g., officials of public tourism organizations, heads of tourism associations) were recruited for the interviews to obtain in-depth information.  

Point 6: Why October and November? Is that intentional?

Response 6: We select this period as the social distancing restrictions were eased at that time at the examined case study site. As such, we are able to contact in person and interview the respondents.

Point 7: What kind of questions were asked? The author could also share the questions in the appendices  

Response 7: Yes, we have added the questions at the end of the introduction section.

Point 8: Findings and analysis/Discussion- The findings presented were very generic (almost sounds like it is applicable to any city). For instance decline in customers, service and staff all this are happening worldwide. Table 2 shows the measures that limit the spread of diseases which were also generic. You don't need an interview to know this. 

Response 8: Yes, we have restructured the discussion section to reflect the vulnerability of the local tourism system across different sectors. We also further discussed how a local tourism system could rebuild its resilience and recovery in the post-pandemic. Hopefully the theoretical and practical implications are useful for other destinations.

Point 9: The paper will read better if the author is able to provide more information specifically applicable to Can tho city instead of generic findings.

Response 9: Yes, we have improved this by adding more information about the research implications, specifically for the case study (Can Tho city).

Many thanks again.

Kind regards

Authors

Round 2

Reviewer 1 Report

Congratulations

Reviewer 2 Report

The authors have answered all the questions posted. I am happy with the ammendments